EMBO
Molecular Medicine

# Molecular cause and functional impact of altered synaptic lipid signaling due to a *prg-1* gene SNP

Johannes Vogt[1],[*],[†], Jenq-Wei Yang[2],[†], Arian Mobascher[3],[†], Jin Cheng[1], Yunbo Li[1], Xingfeng Liu[1], Jan Baumgart[1], Carine Thalman[1], Sergei Kirischuk[2], Petr Unichenko[2], Guilherme Horta[1], Konstantin Radyushkin[4], Albrecht Stroh[1], Sebastian Richers[1], Nassim Sahragard[1], Ute Distler[4],[5], Stefan Tenzer[5], Lianyong Qiao[1], Klaus Lieb[3], Oliver Tüscher[3], Harald Binder[6], Nerea Ferreiros[7], Irmgard Tegeder[7], Andrew J Morris[8], Sergiu Gropa[9], Peter Nürnberg[10], Mohammad R Toliat[10], Georg Winterer[10],[†], Heiko J Luhmann[2],[†], Jisen Huai[1],[†] & Robert Nitsch[1],[†],[**]

## Abstract

Loss of plasticity-related gene 1 (PRG-1), which regulates synaptic phospholipid signaling, leads to hyperexcitability via increased glutamate release altering excitation/inhibition (E/I) balance in cortical networks. A recently reported SNP in *prg-1* (R345T/mutPRG-1) affects ~5 million European and US citizens in a monoallelic variant. Our studies show that this mutation leads to a loss-of-PRG-1 function at the synapse due to its inability to control lysophosphatidic acid (LPA) levels via a cellular uptake mechanism which appears to depend on proper glycosylation altered by this SNP. PRG-1[+/−] mice, which are animal correlates of human PRG-1[+/mut] carriers, showed an altered cortical network function and stress-related behavioral changes indicating altered resilience against psychiatric disorders. These could be reversed by modulation of phospholipid signaling via pharmacological inhibition of the LPA-synthesizing molecule autotaxin. In line, EEG recordings in a human population-based cohort revealed an E/I balance shift in monoallelic mutPRG-1 carriers and an impaired sensory gating, which is regarded as an endophenotype of stress-related mental disorders. Intervention into bioactive lipid signaling is thus a promising strategy to interfere with glutamate-dependent symptoms in psychiatric diseases.

**Keywords** bioactive phospholipids; cortical network; PRG-1; psychiatric disorders; synapse
**Subject Categories** Genetics, Gene Therapy & Genetic Disease; Neuroscience

See also: **B Stutz & TL Horvath** (January 2016)

## Introduction

Accurate synaptic transmission is a fundamental requirement for normal brain function (Turrigiano, 2011), and signaling alterations at the excitatory synapse have been related to psychiatric disorders such as schizophrenia (Harrison & Weinberger, 2005; Coyle, 2006; Belforte *et al*, 2010). Recent research shows that bioactive lipid signaling and protein–lipid interaction play major roles in all steps of endo- and exocytosis processes, including synaptic vesicle cycling (Di Paolo *et al*, 2004). Plasticity-related gene 1 (PRG-1, also assigned as LPPR4), a neuron-specific molecule in the brain, is an important postsynaptic control element of this pathway (Trimbuch *et al*, 2009). Previously, we demonstrated that the absence of PRG-1 (Brauer *et al*, 2003), which is involved in synaptic phospholipid signaling, leads to hippocampal hyperexcitability via increased glutamate release at the synapse (Trimbuch *et al*, 2009). Recently, a single nucleotide polymorphism (SNP, rs138327459, NHLBI Exome Sequencing Project https://esp.gs.washington.edu/drupal/) was detected in humans resulting in an arginine (R) to threonine (T) exchange at position 345 of the amino acid chain (mouse

1  Institute for Microscopic Anatomy and Neurobiology, University Medical Center, Johannes Gutenberg-University, Mainz, Germany
2  Institute for Physiology and Pathophysiology, University Medical Center, Johannes Gutenberg-University, Mainz, Germany
3  Department of Psychiatry and Psychotherapy, University Medical Center, Johannes Gutenberg-University Mainz, Mainz, Germany
4  Focus Program Translational Neuroscience (FTN), University Medical Center, Johannes Gutenberg-University, Mainz, Germany
5  Institute for Immunology, University Medical Center, Johannes Gutenberg-University Mainz, Mainz, Germany
6  Institute of Medical Biostatistics, Epidemiology and Informatics, University Medical Center, Johannes Gutenberg-University Mainz, Mainz, Germany
7  Institute of Clinical Pharmacology Goethe-University Hospital, Frankfurt am Main, Germany
8  Division of Cardiovascular Medicine, Gill Heart Institute, University of Kentucky, Lexington, KY, USA
9  Department of Neurology, University Medical Center, Johannes Gutenberg-University Mainz, Mainz, Germany
10 Cologne Center for Genomics (CCG), University of Cologne, Cologne, Germany
    *Corresponding author. Tel: +49 6131 17 8091/8072; E-mail: johannes.vogt@unimedizin-mainz.de
    **Corresponding author. Tel: +49 6131 17 8091/8072; E-mail: robert.nitsch@unimedizin-mainz.de
    †These authors contributed equally to this work

homolog located at position 346). Our electrophysiological studies using re-expression of PRG-1$^{R346T}$ in PRG-1$^{-/-}$ mice by *in utero* electroporation revealed that this mutation results in a loss-of-PRG-1 function at the synapse. On a molecular level, we could show that PRG-1$^{R346T}$ lacked the ability to support uptake of lysophosphatidic acid (LPA) into intracellular compartments due to altered O-glycosylation of S347 next to the SNP site, while in-depth quantitative analysis revealed no role for serine phosphorylation at this position. PRG-1 heterozygous mice which are an animal correlate for human monoallelic PRG-1$^{+/mut}$ carriers showed altered cortical information processing and stress-related behavioral deficits indicative for mental disorders. Using specific inhibitors of phospholipid synthesis, we could show that modulation of bioactive phospholipid levels upstream of PRG-1 reverted cortical network function and behavior toward wild-type (wt) levels. In line with experimental data, electrophysiological assessment using the P50 sensory gating auditory paradigm (which corresponds to the pre-pulse inhibition (PPI) tested in mice) revealed an altered sensory gating in monoallelic R345T PRG-1 carriers which were identified among 1,811 human volunteers in a population-based cohort. Since similar alterations of cortical excitability and sensory gating have been described as an endophenotype of schizophrenia and stress-related disorders (Turetsky *et al*, 2007; Javitt *et al*, 2008), our results indicate a novel therapeutic strategy targeting synaptic bioactive lipid signaling in altered cortical information processing related to mental disorders.

# Results

### PRG-1$^{R346T}$ is a loss-of-function mutation

PRG-1 was shown to be involved in internalization of lysophosphatidic acid (LPA) to intracellular postsynaptic compartments (Trimbuch *et al*, 2009), thereby controlling LPA levels in the synaptic cleft. To understand the molecular consequences of the human SNP resulting in arginine (R) to threonine (T) exchange at position 345 in the amino acid chain of PRG-1, we established HEK cell lines with stable expression of the mouse homolog of this SNP, PRG-1$^{R346T}$, and found similar membrane localization when compared to wild-type PRG-1 (wtPRG-1, Fig 1A). After application of fluorescence labeled LPA (TopFluor (TF)-LPA) and removal of surface bound TF-LPA using 0.001% SDS, PRG-1$^{R346T}$-expressing cells displayed a lower fluorescence signal in intracellular compartments (Fig 1B). This finding was quantified by FACS analysis (example shown in Fig 1C) revealing a significantly reduced capacity of PRG-1$^{R346T}$-expressing cells to internalize TF-LPA when compared to wtPRG-1 (Fig 1D). To unambiguously prove the internalization of LPA, we applied a chemically modified, unnatural LPA (C17-LPA) on wtPRG-1-expressing cells which showed a statistically significant internalization of C17-LPA (Fig 1E) and the intracellular presence of its metabolite C17-MAG (Fig 1F) when compared to HEK cells which lack any PRG-1 expression. In line with previous results, PRG-1$^{R346T}$-expressing cells did not show a significant increase of intracellular C17 LPA nor of its metabolite C17-MAG providing further evidence that PRG-1$^{R346T}$ lacks the capacity to mediate internalization of LPA into cells (Fig 1E and F).

### PRG-1$^{R346T}$ alters O-glycosylation, but not phosphorylation of neighboring S347

Since PRG-1$^{R346T}$ was properly expressed at the membrane, we hypothesized a posttranslational modification being involved in the loss of function in TF-LPA internalization. Interestingly, the SNP changing R345 (arginine) to 345T (threonine) is located in close proximity to a frequently reported phosphorylation site (Munton *et al*, 2007; Trinidad *et al*, 2008; Huttlin *et al*, 2010; Wisniewski *et al*, 2010; Goswami *et al*, 2012). Therefore, we addressed the question to what extent the neighboring S (serine) at position 346 in humans and at position 347 in mouse is phosphorylated under wild-type conditions, and whether this phosphorylation is eventually altered due to the mutation of R to T, which alters typical phosphorylation motifs (Pearson & Kemp, 1991). Using quantitative mass spectrometry analysis and isotopically labeled peptides corresponding to the peptide containing S347, we detected that, although described as a potential activity-related phosphorylation site (Munton *et al*, 2007), S347 neighboring the SNP site showed only minor phosphorylation (4.73% of total PRG-1, Fig 1G). In addition, in PRG-1$^{R346T}$, we could neither detect any relevant change in phosphorylation of S347 (3.33% of total PRG-1, Fig 1G), nor any phosphorylation of the T at position 346. To further assess whether S347 phosphorylation is a potential relevant cause for the loss of LPA uptake of PRG-1$^{R346T}$, we established HEK cell lines with a stable expression of PRG-1$^{S347A}$ and PRG-1$^{S347D}$, which either mimic the non-phosphorylated (S347A) status or the phosphorylated status (S347D) of S347. Both cell lines lacked the capacity to significantly internalize TF-LPA above baseline levels (Fig 1H), strongly arguing against an important role for phosphorylation explaining the loss of function of PRG-1$^{R346T}$ in TF-LPA internalization, however, pointing to the need of S347 at this position. Another intracellular posttranslational modification, critically important for brain function is O-GlcNAC glycosylation (Rexach *et al*, 2008), which occurs at S and T residues and has a complex interplay with phosphorylation (Tarrant *et al*, 2012). Using immunoprecipitation studies and specific antibodies, we analyzed O-GlcNAC glycosylation in wtPRG-1 in comparison with PRG-1$^{R346T}$ and detected significantly reduced glycosylation levels in PRG-1$^{R346T}$-expressing cells (Fig 1I and J), indicating an interference of the R346T mutation with its neighboring S347 O-glycosylation site. Since neither A nor D is a target for O-GlcNAC glycosylation and the cell lines expressing PRG-1$^{S347A}$ and PRG-1$^{S347D}$ lacked the capacity of internalizing TF-LPA (Fig 1H), these data point to the necessity of an O-glycosylation at S for proper PRG-1 function, which is altered in PRG-1$^{R346T}$.

In sum, these data on posttranslational modifications provide evidence for the fact that not changes in phosphorylation, but proper O-glycosylation of S at position 346 in humans interfered by the R345T SNP is crucial for PRG-1 function.

### PRG-1$^{R346T}$ is a loss-of-function mutation at the synapse

To analyze potential functional consequences of the human SNP in neurons, we moved on in assessing PRG-1$^{R346T}$ function at the synapse and performed whole-cell patch-clamp recordings from layer IV spiny stellate neurons in the S1BF of PRG-1$^{-/-}$ animals that re-expressed PRG-1$^{R346T}$ delivered by *in utero* electroporation (Fig 2A). This re-expression resulted in proper localization of

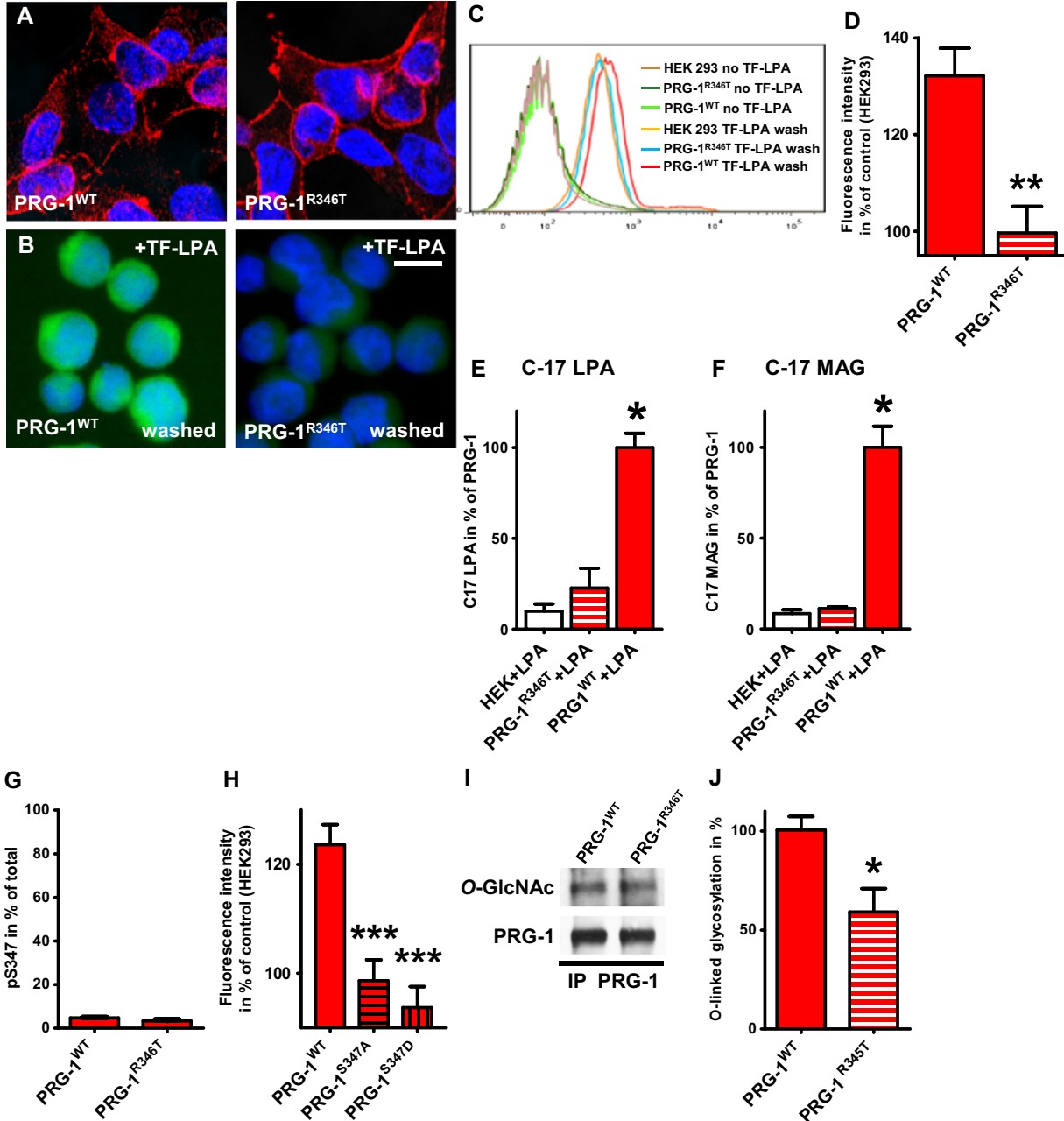

**Figure 1.  PRG-1$^{R346T}$: loss-of-function mutation reduced LPA internalization due to altered O-glycosylation.**

A    PRG-1 and PRG-1$^{R346T}$ are expressed on HEK293 cell membranes.

B    Conventional fluorescence images of transfected and TopFluor (TF)-LPA-stimulated cells after washing with 0.001% SDS. Scale bar: 10 μm.

C, D    FACS analysis of internalized TF-LPA of PRG-1- and PRG-1$^{R346T}$-transfected cells (*n* = 7 each, **\*\****P* = 0.0015; *t*-test).

E, F    Incubation with C-17 LPA and subsequent washing as described above, led to a significant increase of TF-LPA in PRG-1-expressing cells but not in PRG-1$^{R346T}$-expressing cells. In line, C-17 MG, a degradation product of C-17 LPA, was significantly increased in PRG-1-expressing cells but not in control or PRG-1$^{R346T}$-expressing cells (*n* = 4 each; Kruskal–Wallis test (for C-17 LPA *P* = 0.0066; for C-17 MAG *P* = 0.0048) with Dunn's multiple comparison test, \**P* < 0.05 for C-17 LPA and C-17 MAG).

G    Quantitative assessment of S347 phosphorylation using isotopically labeled peptides and mass spectrometry of purified wild-type PRG-1 and PRG-1$^{R346T}$ from stable transfected HEK293 cells revealed low total phosphorylation levels for both protein variants (*n* = 3 measurements).

H    TF-LPA uptake as analyzed by FACS was significantly decreased in S347A and S347D mutant PRG-1 (*n* = 12 per condition, \*\*\**P* < 0.0001; one-way ANOVA with Bonferroni's multiple comparison test).

I    Western blot (WB) of IP using a specific PRG-1 antibody revealed decreased levels of O-GlcNAc in PRG-1$^{R346T}$ when compared to wild-type PRG-1.

J    Densitometric analysis of *n* = 7 WB from IP of PRG-1- and PRG-1$^{R346T}$-expressing cells revealed a significantly decreased O-GlcNAc amount in PRG-1$^{R346T}$ (unpaired *t*-test, \**P* = 0.0107).

Data information: All experiments were done in parallel. Bar diagrams represent mean ± SEM. \**P* < 0.05, \*\**P* < 0.01, \*\*\**P* < 0.001.

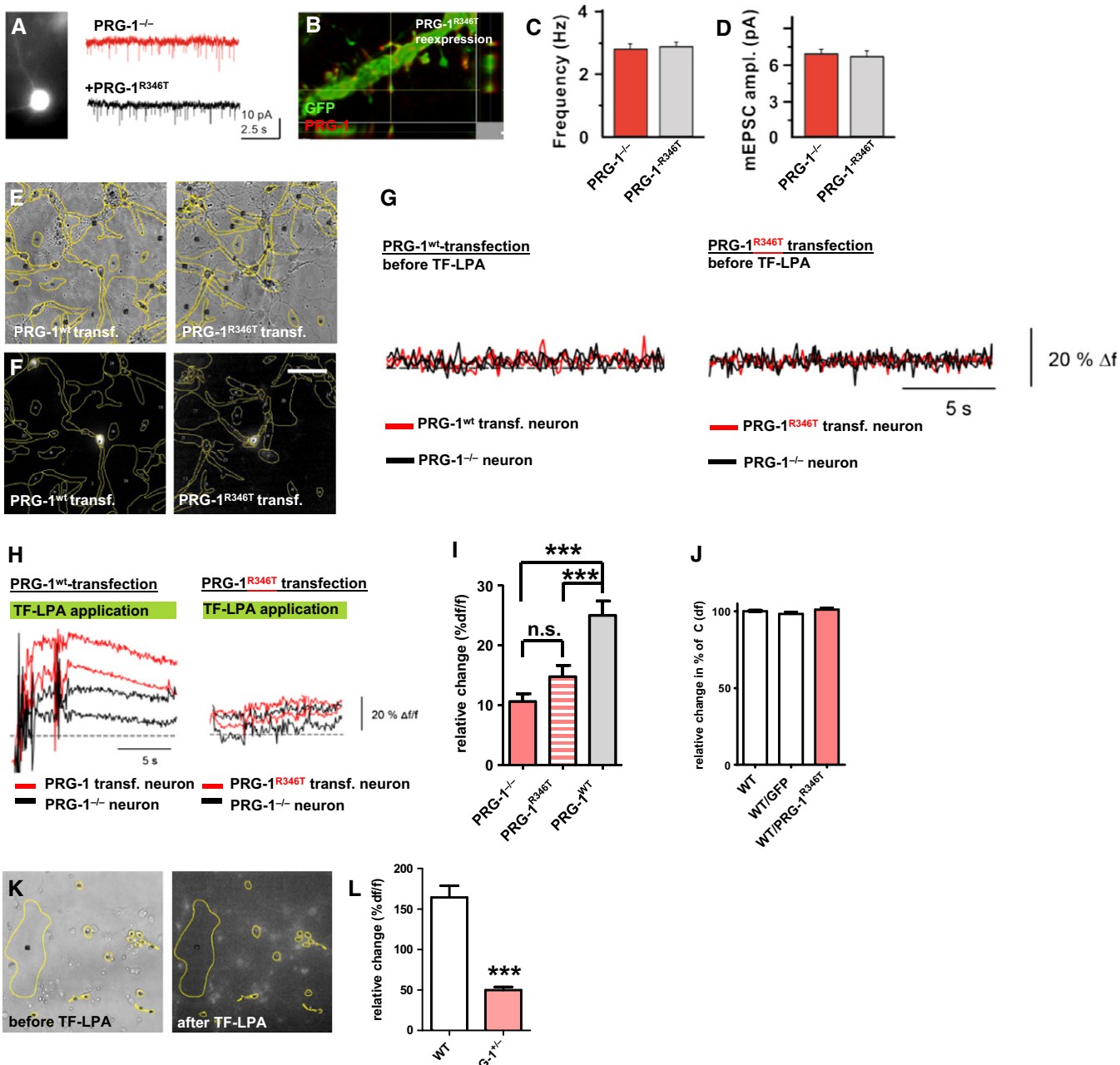

**Figure 2.  PRG-1$^{R346T}$ fails to rescue PRG-1$^{-/-}$ phenotype and to induce significant TF-LPA uptake.**

A, B    *In utero* electroporation (IUE) of PRG-1$^{R346T}$ lead to membrane expression of this construct.

C, D    *In vitro* whole-cell patch-clamp recordings from PRG-1$^{-/-}$ layer IV spiny stellate neurons (S1BF) and of spiny stellate neurons reconstituted with PRG-1$^{R346T}$ by IUE. Original traces are shown in (A) (*n* = 11 PRG-1$^{-/-}$ neurons and 11 PRG-1$^{R346T}$-reconstituted neurons from 4 mice; *t*-test).

E    Bright field image and regions of interest (ROIs, yellow) of primary neuronal cultures containing PRG-1$^{-/-}$ neurons.

F    Detection of transfected neurons via a co-transfected IRES-GFP cassette by fluorescent illumination. Scale bar: 50 μm.

G    Baseline fluorescence (calculated as Δf/f) of transfected and non-transfected neurons prior to TF-LPA stimulation.

H    PRG-1$^{wt}$-transfected neurons displayed a clear increase in fluorescence intensity after TF-LPA application (left), while PRG-1$^{R346T}$-transfected neurons were not different from non-transfected PRG-1$^{-/-}$ neurons (right).

I    Fluorescence intensity analysis of the ROIs revealed significant TF-LPA uptake in PRG-1$^{wt}$-transfected neurons but not in PRG-1$^{R346T}$-transfected neurons (*n* = 25 neurons per condition; repeated-measures ANOVA with Bonferroni *post hoc* correction, ***$P$ < 0.0001).

J    Expression of GFP or of PRG-1$^{R346T}$ in wild-type neurons did not alter TF-LPA uptake when compared to non-transfected neurons (*n* = 57 wild-type, 10 GFP-transfected and 10 PRG-1$^{R346T}$-transfected neurons; Kruskal–Wallis test with Dunn's multiple comparison test).

K    Bright field image and fluorescence image overview showing TF-LPA in neurons with ROIs (yellow) of a primary neuronal culture.

L    Fluorescence analysis revealed a significantly lower TF-LPA uptake in PRG-1$^{+/-}$ neurons when compared to wild-type neurons (*n* = 15 neurons per group; unpaired *t*-test, ***$P$ < 0.0001).

Data information: Bar diagrams represent mean ± SEM.

PRG-1[R346T] protein at the spine head membrane, excluding an impact of this mutation in protein targeting (Fig 2B). While re-expression of wtPRG-1 in PRG-1-deficient neurons was reported to rescue the frequency of miniature excitatory postsynaptic currents (mEPSC) to wild-type levels in hippocampal CA1 pyramidal cells (Trimbuch *et al*, 2009), PRG-1[R346T], albeit present at the proper dendritic localization, failed to achieve this functional rescue. This is consistent with a loss of function of PRG-1[R346T], presumably due to its inability to support LPA internalization (Fig 2A and C).

To directly assess the functional consequences of PRG-1[R346T] in neurons in terms of their capacity to internalize LPA (Trimbuch *et al*, 2009), we used primary cortical neurons obtained from PRG-1[−/−] animals either transfected with a *prg-1*[wt] or with a *prg-1*[R346T] expressing construct (Fig 2E and F) and measured internalization of LPA which was tagged by a fluorescence label (TopFluor, TF-LPA) in a live imaging mode. TF-LPA internalization capacity of either PRG-1[wt] or PRG-1[R346T] was compared to PRG-1-deficient neurons in the same experiment (see also Fig 1G for baseline levels). To exclude bias by the transfection procedure, we assessed the TF-LPA uptake capacity of GFP-transfected PRG-1[−/−] neurons finding no difference to the non-transfected neurons (Fig EV1). While expression of PRG-1[wt] in neurons induced a significant increase in TF-LPA internalization, PRG-1[R346T] expression failed to do so (Fig 2H and I). To rule out a dominant-negative effect of PRG-1[R346T], we used the same approach and transfected cortical neurons obtained from wild-type mice either with a *prg-1*[R346T] or a control GFP construct and found no alteration on TF-LPA-internalization in the transfected neurons (Fig 2J). These studies confirmed that PRG-1[R346T] was a loss-of-function mutation in a neuronal context. Finally, since the reported human SNP was only detected in monoallelic carriers, we analyzed the impact of a monoallelic functional loss in cortical neurons obtained from PRG-1[+/−] animals and found that already monoallelic PRG-1 loss was sufficient to significantly decrease TF-LPA-internalization when compared to wild-type neurons (Fig 2K and L).

**Altered phospholipid modulation due to monoallelic loss of synaptic PRG-1 affects cortical E/I balance**

In order to characterize the role of PRG-1 in cortical networks, we assessed expression of PRG-1 in the cerebral cortex and found strong expression in layer IV of the mouse somatosensory barrel field cortex (S1BF; Fig 3A) which was restricted to glutamatergic neurons (Fig EV2). Analysis of subcellular localization revealed that PRG-1 was expressed at dendritic spines (Fig 3B and C). Since heterozygous deletion of PRG-1, leaving only one functional *prg-1* allele, results in a linear reduction of protein expression of approximately 50% (Trimbuch *et al*, 2009), and heterozygous PRG-1-deficiency significantly reduced the functional capacity of PRG-1[+/−] neurons to internalize TF-LPA in a similar range (Fig 2K and L), PRG-1[+/−] animals are to be assumed as an animal correlate of human monoallelic PRG-1[R345T] carriers who only have one functional *prg-1* allele not affected by the mutation. To understand implications of such a reduced synaptic PRG-1 function in intracortical information processing, we performed multichannel extracellular recordings in the S1BF cortex of wt and PRG-1[+/−] mice *in vivo* (Fig 3D). Spontaneous activity analysis in PRG-1[+/−] mice showed a significant prolongation of multiunit activity (MUA) burst duration when compared to wt mice (Fig 3E and F). This finding points to a change in the cellular E/I

balance toward excitation within cortical microcircuitries which has been related to severe behavioral deficits (Yizhar *et al*, 2011).

**Inhibition of LPA synthesis reverts altered cortical information processing in monoallelic PRG-1-deficient mice**

To test for involvement of phospholipids at the cortical network level, we analyzed the effect of inhibiting the LPA-synthesizing molecule autotaxin (ATX) which acts upstream of the LPA-LPA$_2$/PRG-1 axis (Moolenaar & Perrakis, 2011). HA-130, a recently described specific inhibitor of ATX (Albers *et al*, 2010), significantly decreased spontaneous postsynaptic currents (sPSC), which are the sum of the overall excitatory and inhibitory input reflecting the total E/I balance (Fig EV3). In line with these findings *ex vivo*, concentrations as low as 1 μM, HA 130, applied into the cortex of living PRG-1[+/−] mice, significantly decreased the number of multiunit activity (MUA) bursts (Fig 3G).

In addition to analysis of spontaneous cortical network activity, we assessed pre-attentive cortical information processing and measured sensory gating in a double-pulse whisker stimulation model (Fig 4A). Local field potentials (recorded in the whisker-specific cortical barrel) in response to repetitive single whisker stimulation elicited larger amplitudes to second stimulus in PRG-1[+/−] when compared to wt (S1; S2; ISI = 500 ms; Fig 4B). Larger S2 amplitudes altered S1–S2 values and resulted in a statistically significant increase of the S2/S1 ratio (Fig 4C and D). These findings indicate that already loss of one functional allele of *prg-1* as present in mutPRG-1 carriers, and thus a reduction of about 50% of functional PRG-1 at the synapse, causes an apparent E/I imbalance in cortical networks and an altered sensory gating. To prove whether phospholipid modulation is able to directly affect cortical information processing, we applied HA-130 in the double-pulse whisker stimulation model (Fig 4E). This inhibition of LPA synthesis significantly restored sensory gating in animals with monoallelic PRG-1-deficiency as shown by reduction of S2 values (when compared to S1 values) and a reduced S2/S1 ratio (Fig 4F and G).

**Pharmacological intervention into monoallelic PRG-1 deficiency reverts behavioral deficits characteristic for mental disorders**

To answer the question whether observed alterations in excitatory synaptic function and cortical network activity result in a behavioral phenotype, we assessed monoallelic PRG-1-deficient animals (PRG-1[+/−]). Using a three chambered box paradigm (Radyushkin *et al*, 2009), we found a significantly reduced social interaction index in PRG-1[+/−] mice (Figs 5A and EV4A). We further assessed PRG-1[+/−] mice under normal conditions and after application of environmental stress, known to be a risk factor for mental disorders (van Winkel *et al*, 2008). While PRG-1[+/−] mice displayed no differences under normal conditions in tail suspension tests, following acute stress, PRG-1[+/−] mice showed a significantly higher immobility (Fig 5B), indicative of alterations of resilient behaviors resembling aspects of stress-related mental disorders (Cryan *et al*, 2005; Shen *et al*, 2008; Inta *et al*, 2010).

Since reduced social interaction and altered behavior in tail suspension test reflect symptoms of stress-related mental disorders and alterations in resilient behaviors, we assessed pre-pulse inhibition (PPI) in these animals. PPI is highly conserved among

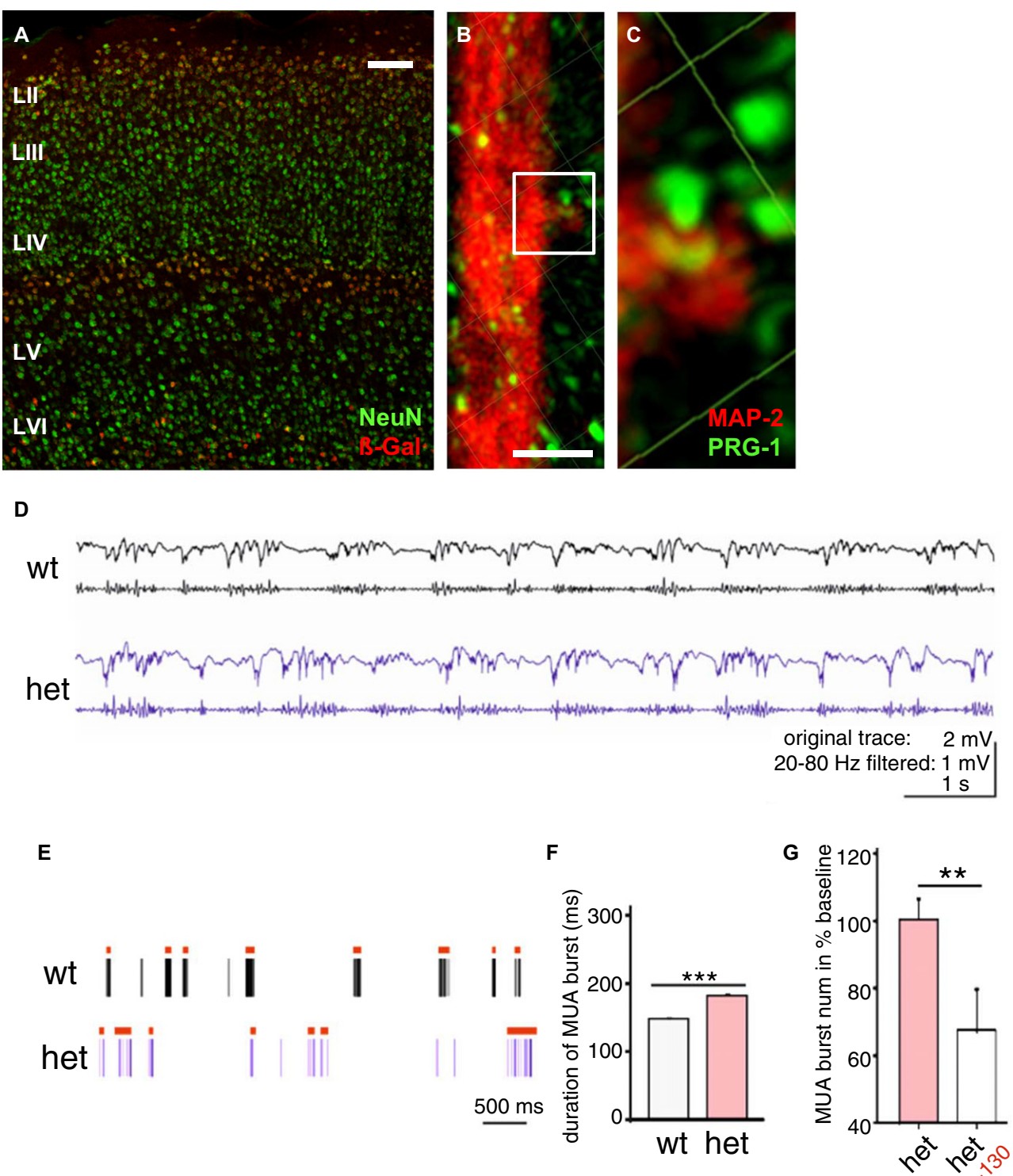

**Figure 3. PRG-1 expression in the somatosensory barrel field cortex (S1BF) and *in vivo* electrophysiological assessment.**

A　　PRG-1 is strongly expressed in layer IV neurons of the S1BF as shown by the β-Gal reporter. Scale bar: 100 μm.

B, C　PRG-1 is localized on dendrites (B) and on MAP-2-positive spines (C). Scale bar: 5 μm.

D–F　Cortical network assessment in PRG-1$^{+/-}$ mice as revealed by spontaneous MUA activity (spikes: lines; bursts: bars) analyzed in wt (11,379 bursts; five mice) and PRG-1$^{+/-}$ (17,919 bursts; eight mice; unpaired *t*-test, ***$P < 0.0001$).

G　　MUA burst numbers in PRG-1$^{+/-}$ mice before and after application of the ATX inhibitor HA-130 ($n = 9$, paired *t*-test, **$P = 0.0053$).

Data information: Bar diagrams represent mean ± SEM.

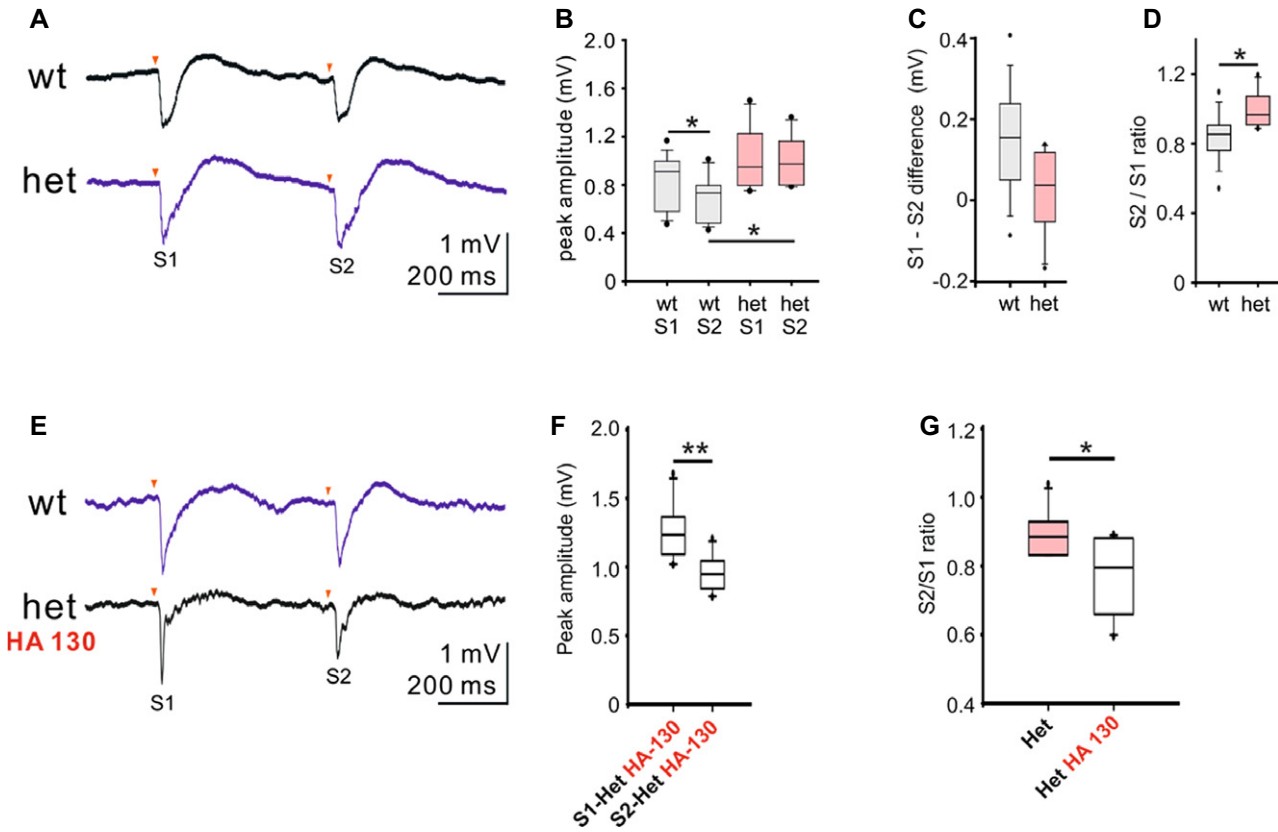

**Figure 4.  PRG-1$^{+/-}$ mice display an altered sensory gating which is rescued by ATX inhibition.**

A–D  Paired-pulse responses (500 ms ISI) to single whisker mechanical stimulation (trace show average of 30 responses) (A) and analysis of peak amplitudes (B), S1-S2 difference (C), and S2/S1 ratio (D) (Wilcoxon matched-pair signed rank test for wtS1 vs. wtS2, *$P$ = 0.0137 and unpaired $t$-test for wtS2 vs. hetS2. *$P$ = 0.0118 for B; Mann-Whitney $U$-test was used for C and D, *$P$ = 0.0446; $n$ = 10 WT animals and 6 PRG-1 het animals.)

E  Paired-pulse responses before and after application of the ATX inhibitor HA-130.

F, G  Peak amplitudes and S2/S1-ratio before and after ATX inhibition by HA-130 ($n$ = 6 animals). Data comparing paired-pulse responses before and after application of HA-130 were assessed by a one-tailed paired $t$-test (**$P$ = 0.0093, *$P$ = 0.0134).

Data information: Box plots represent the 25$^{th}$ and 75$^{th}$ percentile. The middle line shows the 50$^{th}$ percentile (median). Whiskers indicate the 10$^{th}$ and 90$^{th}$ percentiles.

vertebrates, is indicative for altered mental function, and is one of few paradigms in which humans and rodents are tested in similar fashions (Davis, 1984). Based on this, we studied PPI of the startle response in PRG-1$^{+/-}$ (het) and wild-type (wt) mice finding that PRG-1$^{+/-}$ animals showed a significant decrease of PPI in comparison with wt littermates (Fig 5C). To test whether pharmacological treatment is able to revert these behavioral abnormalities indicative for mental disorders, we tested an *in vivo* inhibitor of the LPA-synthesizing enzyme autotaxin (PF8380) which has a nanomolar potency and can be applied orally or by injection to the periphery (Gierse *et al*, 2010). Intraperitoneal injection of PF8380 significantly diminished LPA levels in the cerebrospinal fluid (Fig EV4B) and effectively restored the observed PPI deficit in freely moving PRG-1$^{+/-}$ mice to wild-type levels (Fig 5C).

### PRG-1$^{R345T}$: a human SNP associated with an endophenotype for mental disorders

Analysis of PRG-1 expression in the human cortex revealed strong neuronal PRG-1 signals and a clear expression in layer IV neurons

(Fig 6A). Using specific markers like GAD67, parvalbumin, and calretinin, we found specific PRG-1 expression in glutamatergic neurons but not in interneurons (Fig EV5A–C). Further subcellular analysis revealed dendritic co-localization of PRG-1 with the postsynaptic density marker PSD95 (Fig 6B) confirming specific postsynaptic PRG-1 expression in glutamatergic synapses of excitatory neurons as described in the rodent brain (Trimbuch *et al*, 2009).

To prove for the relevance of PRG-1 and its loss-of-function SNP in humans, we analyzed a well-characterized sample of 1,811 healthy adult German subjects (Brinkmeyer *et al*, 2011; Lindenberg *et al*, 2011; Quednow *et al*, 2012) (see Appendix Table S1 for sociodemographic variables) where we detected the PRG-1$^{R345T}$ (mutPRG-1) variant at a heterozygous frequency of 0.66% (12 carriers). EEG analysis of PRG-1$^{+/mut}$ carriers revealed alterations of the P50 event-related cortical potential (ERP) elicited by an auditory double-stimulus paradigm (stimulus 1 = S1, stimulus 2 = S2), a measure of pre-attentive stimulus processing/sensory gating and an endophenotype for mental disorders (Turetsky *et al*, 2007; Javitt *et al*, 2008). PRG-1$^{+/mut}$ carriers showed larger P50 amplitudes in response to the second stimulus (S2), a finding pointing to an

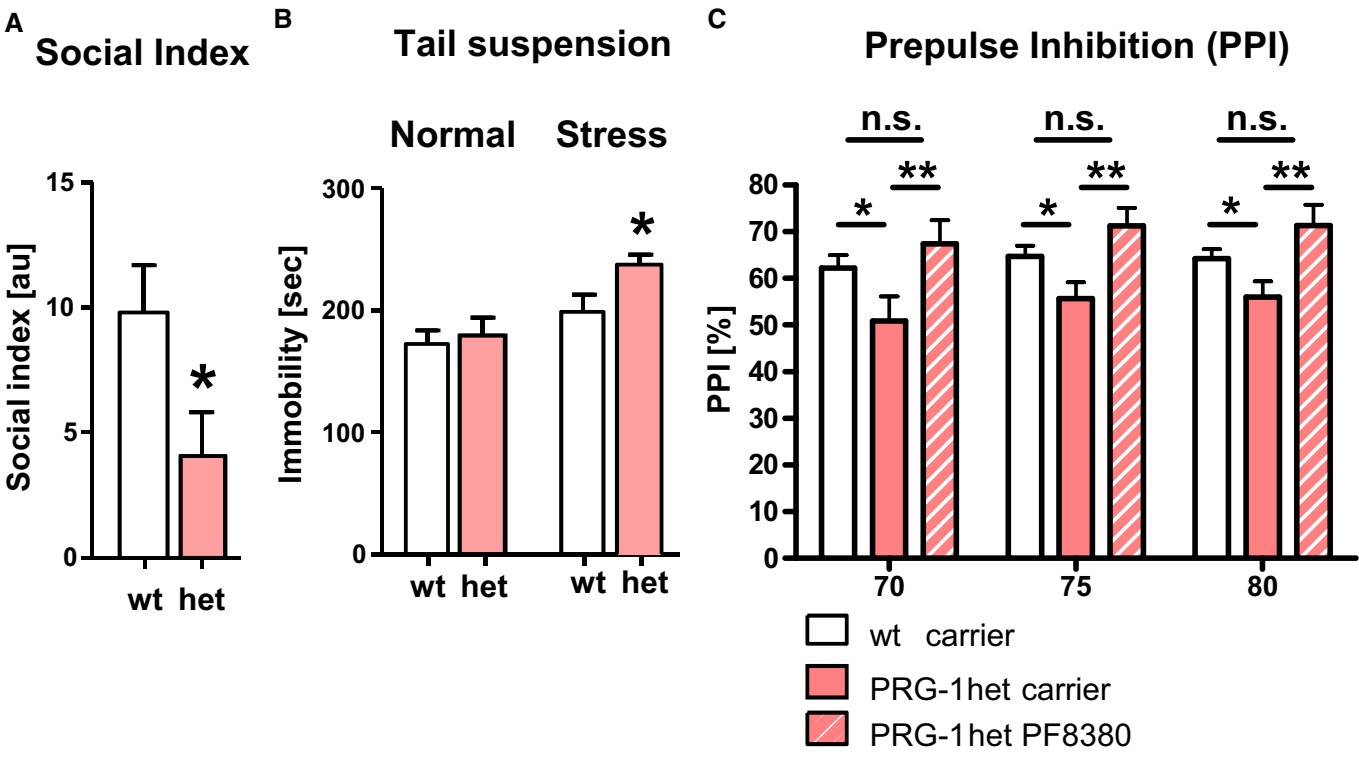

**Figure 5. PRG-1$^{+/-}$ mice display an endophenotype typical for psychiatric disorders which was rescued by ATX inhibition.**

A The social interaction index was significantly reduced in PRG-1$^{+/-}$ (het) mice ($n$ = 12 wild-type and 14 het mice; unpaired $t$-test, *$P$ = 0.035).

B Tail suspension test under normal conditions and after exposure to acute restrain stress ($n$ = 14 PRG-1 het and 12 wt animals; unpaired $t$-test, *$P$ = 0.0214).

C PRG-1$^{+/-}$ mice displayed a robust deficit in PPI at all pre-pulse stimuli (*$P$ = 0.0178). ATX inhibition via PF8380 treatment significantly rescued the PPI deficit in PRG-1$^{+/-}$ mice to a level which was not different to the PPI measured in wt mice ($n$ = 19 wt animals, 12 PRG-1 het animals, and 10 PRG-1 het animals + PF8380; two-way ANOVA with Bonferroni *post hoc*, **$P$ = 0.0068).

Data information: Bar diagrams represent mean ± SEM.

excitation–inhibition imbalance and deficient "gating out" of redundant information (Chang *et al*, 2011) (Fig 6C). S1 reduction and S2 increase/diminished suppression may both contribute to sensory gating abnormalities in mental disorders but have been considered to reflect distinct/partially overlapping processes with a non-identical genetic architecture (Javitt *et al*, 2008; Chang *et al*, 2011).

Since PRG-1 is expressed in human excitatory neurons in the somatosensory cortex (Figs 6A and B and EV5A–C), and PRG-1-deficient mice (PRG-1$^{-/-}$) display hippocampal hyperexcitability (Trimbuch *et al*, 2009), the electrophysiological phenotype of human PRG-1$^{+/mut}$ carriers suggested a functional alteration of mutPRG-1. This apparently lead to an increased synaptic glutamate release at excitatory neurons thus inducing a shift in the E/I balance toward excitation (Yizhar *et al*, 2011) and an altered function of cortical networks as observed in the pathophysiology of mental disorders (Coyle, 2006; Belforte *et al*, 2010; Javitt *et al*, 2011).

## Discussion

Bioactive lipid signaling at the synapse has only recently been shown to be involved in the control of glutamatergic transmission. Evidence from studies in hippocampal neurons highlights lysophosphatidic acid (LPA) as an active substance stimulating presynaptic

LPA receptors which in turn control glutamate release from the presynaptic terminal (Trimbuch *et al*, 2009). Synaptic LPA appears to be under tight control of the postsynaptic lipid phosphatase-like molecule PRG-1 (Brauer *et al*, 2003; McDermott *et al*, 2004), while lack of PRG-1 and thus insufficient control of synaptic LPA action result in overexcitation and eventually hippocampal epilepsy (Trimbuch *et al*, 2009). We found that a single nucleotide polymorphism (SNP) in the human *prg-1* gene resulting in a single amino acid exchange (R345T in humans, mutPRG-1; homolog in mouse: R346T) turned out to be a loss-of-function mutation of PRG-1. This SNP with a monoallelic frequency of ~0.6% affects ~3.5 million European and ~1.5 million US citizens and is homologous to the SNP (rs138327459) reported by the NHLBI Exome Sequencing Project (https://esp.gs.washington.edu/drupal/) in the isoform 2 of PRG-1 (NP_001159724). Here, we analyzed the underlying molecular cause of this loss-of-function SNP and show that loss of function of postsynaptic PRG-1 at the glutamatergic synapse leads to an endophenotype (reduced sensory gating) described in schizophrenia.

Phosphorylation is an important posttranslational modification of synaptic proteins critically influencing their function (Lu & Roche, 2012). Since studies from different laboratories (Munton *et al*, 2007; Trinidad *et al*, 2008; Huttlin *et al*, 2010; Wisniewski *et al*, 2010; Goswami *et al*, 2012) have identified S347 as phosphorylation site in

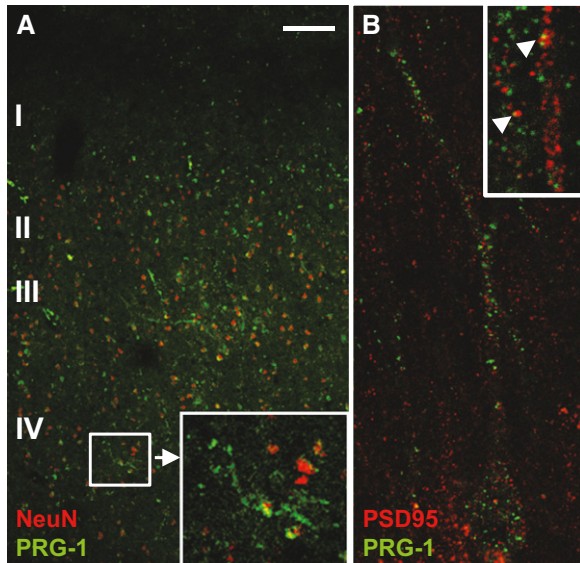

| | prg-1 wt/wt; N=1799 | R345T carriers; N=12 | |
|---|---|---|---|
| | Mean (Std) | Mean (Std) | P-Value |
| S1 amplitude (μV) | 6.95 (1.00) | 7.47 (0.32) | 0.373 |
| S2 amplitude (μV) | 4.47 (0.74) | 5.34 (1.35) | **0.042** |
| S1-S2 difference (μV) | 2.48 (0.86) | 2.13 (1.11) | 0.073 |
| S2/S1 ratio | 0.65 (0.10) | 0.71 (0.15) | 0.361 |
| Main genotype effect (S1 + S2 combined)* | - | - | **0.0006** |

*repeated measures analysis

**Figure 6. PRG-1 R345T: a loss-of-function human mutation induces an endophenotype for mental disorders.**

A  PRG-1 is expressed in NeuN-positive neurons of the somatosensory human cortex. Remark the PRG-1 expression delineating dendritic profiles of NeuN-positive neurons. Insert depicts dendritic PRG-1 expression in a layer IV spiny stellate neuron at higher magnification. Scale bar: 100 μm.

B  PRG-1 and PSD-95 punctae delineate dendritic profiles of a pyramidal neuron. Higher magnification (insert) exemplarily shows PRG-1/PSD95 co-localization (arrow heads).

C  Stratified analysis of *prg-1* genotype on P50 event-related potential in mutPRG-1 carriers.

PRG-1, which together with the preceding R346 forms a typical phosphorylation motif (Pearson & Kemp, 1991), we have analyzed the phosphorylation status of wild-type PRG-1 and PRG-1$^{R346T}$. Interestingly, applying quantitative studies using isotopically labeled peptides, we neither detected a biologically significant amount of S347 phosphorylation in wild-type PRG-1 (wtPRG-1), nor a significant change in the mouse SNP homolog (PRG-1$^{R346T}$). This was corroborated by two mutation studies (PRG-1$^{S347A}$ and PRG-1$^{S347D}$) mimicking the non-phosphorylated and the constitutive phosphorylated form, which both failed to significantly take up TF-LPA pointing to the necessity of S at this position (S347). However, assessment of *O*-GlcNAc glycosylation, another posttranslational modification critical for correct brain function (Cole & Hart, 2001; Tallent *et al*, 2009) occurring at S and T residues and found in a reciprocal interplay with phosphorylation (Cheng & Hart, 2001), revealed significantly reduced glycosylation levels in PRG-1$^{R346T}$ when compared to wtPRG-1. Interestingly, *O*-GlcNAC glycosylation typically occurs in intrinsically disordered protein regions (Trinidad *et al*, 2012) as present in the intracellular C-term of PRG-1 containing S347 and, although consensus motifs of the responsible enzyme GlcNAc-β-Ser/Thr (OGT) have not been set (Moremen *et al*, 2012), arginine (R) located next to *O*-GlcNAC glycosylation sites was described for many proteins (for overview see Lazarus *et al*, 2011). Altogether, our data point to glycosylation as a critical mechanism for PRG-1 function, which was significantly reduced by the PRG-1$^{R346T}$ SNP.

To prove that mutPRG-1 is a loss-of-function mutation also in a neuronal context, we re-expressed the mouse homolog of the human SNP (PRG-1$^{R346T}$) in PRG-1$^{-/-}$ mice and assessed synaptic function using single cell electrophysiology. While it was shown that PRG-1 reconstitution in the hippocampus restored high mEPSC to normal levels (Trimbuch *et al*, 2009), PRG-1$^{R346T}$ re-expression,

although correctly localized at dendritic spines, failed to do so. In line, functional analyses of PRG-1$^{WT}$- or PRG-1$^{R346T}$-reconstituted PRG-1$^{-/-}$ neurons in terms to modulate extracellular LPA showed that PRG-1$^{R346T}$ reconstitution failed to induce significant TF-LPA uptake which was present in PRG-1$^{WT}$-reconstituted neurons. However, since (i) PRG-1$^{R346T}$ reconstitution did not exert a dominant-negative effect on TF-LPA uptake when expressed in WT neurons and (ii) neurons with monoallelic PRG-1 deletion (PRG-1$^{+/-}$) displayed 50% reduction in TF-LPA uptake, which corresponds to the halfway reduction of the PRG-1 protein in PRG-1$^{+/-}$ mice (Trimbuch *et al*, 2009), our data show that heterozygous PRG-1 mice are a valid animal model to study synaptic alterations present in monoallelic PRG-1$^{WT/mut}$ carriers.

Increased glutamate at the synapse is involved in a broad range of diseases and alterations in cortical excitation/inhibition (E/I) balance have been suggested to contribute to the pathophysiology underlying mental disorders (Yizhar *et al*, 2011). In our present study, we could show that control of glutamatergic transmission by bioactive lipid signaling is an important prerequisite for proper cortical information processing. PRG-1 is highly expressed in layer IV neurons of the somatosensory cortex and already 50% reduction of functional active PRG-1 at the glutamatergic synapse caused reduced LPA uptake into postsynaptic neurons, and as a consequence, shifted E/I balance toward a higher excitability of cortical networks leading to pathological cortical information processing as observed in schizophrenia (Coyle, 2006; Belforte *et al*, 2010; Hasan *et al*, 2012a, 2012b; Yizhar *et al*, 2011). This functional impairment leads to decreased social interaction, loss of resilience in tail suspension test after acute stress application, and decreased PPI which are well-established animal correlates of altered mental function (Shen *et al*, 2008; Inta *et al*, 2010). Using PRG-1$^{+/-}$ mice, which resemble

the PRG-1$^{WT/mut}$ situation present in human SNP carriers, we could show for the first time that proper bioactive lipid signaling at the synapse is important for cortical information processing. This is supported by data obtained from EEG analyses in humans showing that P50 event-related potentials, a measure of sensory processing and gating reflecting E/I balance alterations and impaired cortical network function, were significantly altered in monoallelic mutPRG-1 human carriers. In sum, our study thus provides a translational line of evidence showing that alterations in E/I balance and pre-attentive stimulus processing, which are considered biomarkers/endophenotypes of psychiatric disorders (Coyle, 2006; Javitt et al, 2008, 2011) and are present in PRG-1$^{+/mut}$ carriers, result from a reduced PRG-1 action at the synapse, leading to increased LPA levels and loss of proper cortical network function. Our study indicates a role of PRG-1$^{R346T}$ as a potential susceptibility gene for human neuropsychiatric disorders.

Our results provide further support for the notion that cortical excitation/inhibition imbalances contribute to glutamate-related electrophysiological and behavioral abnormalities in psychiatric disorders (e.g., the glutamate hypothesis of schizophrenia). Moreover, they suggest that proper bioactive lipid signaling is important for normal sensory discrimination and learning, which both depend on synaptic homeostatic plasticity controlling accurate E/I balance as a prerequisite for appropriate resilience against stress-related mental dysfunction (Kalisch et al, 2015). Thus, in addition to the known NMDA receptor hypofunction on inhibitory neurons (reduced feed-forward inhibition), alteration of glutamatergic activity on excitatory neurons (increased feed-forward excitation) may contribute to psychiatric diseases (Coyle, 2006; Belforte et al, 2010; Javitt et al, 2011; Yizhar et al, 2011).

Since modulation of the phospholipid signal cascade using LPA synthesis inhibitors acting upstream of PRG-1 was effective in reverting the decreased sensory gating and the altered behavior in PRG-1$^{+/-}$ mice toward wild-type levels, our data provide evidence that E/I balance critically depends on accurate regulation of synaptic LPA levels. This synaptic regulatory mechanism seems to depend on a proper posttranslational modification like glycosylation of the synaptic phospholipid modulator PRG-1 and alterations in this signaling pathway may result in altered cortical information processing.

LPA levels are controlled by its synthesizing enzyme, autotaxin on the one hand (Moolenaar & Perrakis, 2011), and by PRG-1 from the postsynaptic side on the other (Trimbuch et al, 2009). Our previous studies in the hippocampus have shown that affecting this LPA signaling pathway by additional LPA$_2$ receptor deletion rescues the phenotype observed in animals lacking synaptic control of bioactive lipid signaling due to PRG-1 deficiency (Trimbuch et al, 2009). In the present study, we show that an upstream intervention in the LPA-LPA$_2$/PRG-1 signaling axis (achieved by ATX inhibition) rescued the E/I balance in the same way (reducing the higher spontaneous glutamate release and restoring sensory (double-pulse whisker stimulation model) and sensorimotor gating (PPI) in PRG-1$^{+/-}$ mice) as shown in our genetic studies using PRG-1$^{-/-}$// LPA$_2$$^{-/-}$ deficient mice strains.

In sum, pharmacological intervention in the animal model of mutPRG-1 via upstream manipulation of the above-described pathway using available ATX inhibitors was successful in rescuing sensorimotor gating (PPI) at electrophysiological and behavioral

level, known to resemble aspects of mental disorders (Swerdlow et al, 1994; Cryan et al, 2005; Radyushkin et al, 2009). Clearly, further in-depth prospective analysis of PRG-1$^{+/mut}$ carriers and case–control studies in patient cohorts with psychiatric diseases is needed to ultimately design novel treatment approaches based on targeted intervention to synaptic lipid signaling. However, our data suggest that interfering with LPA signaling via inhibitors of autotaxin might thus represent a promising strategy to treat the human phenotype of PRG-1$^{WT/mut}$ or other stress-mediated behavioral alterations and glutamate-related mental disorders (Harrison & Weinberger, 2005; Coyle, 2006; Belforte et al, 2010; Javitt et al, 2011; Hasan et al, 2012b; Kalisch et al, 2015).

# Materials and Methods

### Electrophysiological and behavioral studies in mice

All experiments were conducted in mice on a C57BL/6J background in accordance with the national laws for the use of animals in research and with the European Communities Council Directive 86/609/EEC and approved by the local ethical committee (Landesuntersuchungsamt Rheinland-Pfalz 23.177-07/G 10-1-010 and 23. 177-07/G 12-1-096). Experiments were designed to minimize the number of animals used. PRG-1$^{-/-}$ mice were obtained by genetic modification as described in Trimbuch et al (2009) and backcrossed on a C57Bl/6J background (Jackson) for at least 10 generations. In utero electroporation, recordings in vitro and in vivo were performed following standard protocols and are described in more detail in the Appendix Supplementary Methods.

### LPA uptake assay

HEK293 cells and HEK293 cell lines expressing wild-type PRG-1, PRG-1$^{R346T}$ (corresponding to the human R345T mutation), PRG-1$^{S346A}$, and PRG-1$^{S346D}$ were incubated with TF-LPA [Avanti Polar Lipids, USA (Saunders et al, 2011)] at 1 μM for 5 min at 37°C. After removing TF-LPA from the cell surface by washing the cells (3 × 5 min) with 0.001% SDS in DMEM, cells were assessed by flow cytometry (FACSCanto II, Becton Dickinson, USA) and analyzed by the FlowJo software (Tree Star, USA). TF-LPA uptake was either calculated as changes of internalized TF-LPA when compared to non-transfected HEK293 cells or to PRG-1-expressing cells. In another set of experiments, cells were incubated with non-natural lipid C17-LPA (Avanti Polar Lipids, USA) instead of TF-LPA. C17-LPA and its degradation product C17-MG were subsequently assessed by mass spectrometry. Primary neuronal cultures were prepared and transfected according to standard procedures. TF-LPA stimulation and optical detection of neuronal TF-LPA uptake was performed as described in the Appendix Supplementary Methods.

### Quantitative phosphorylation studies

Isotopically labeled peptides (Heavy Peptide™ AQUA QuantPro) were purchased from Thermo Fisher Scientific (Waltham, MA). For assessing S347 phosphorylation in wt PRG-1 and in PRG-1$^{R346T}$ following heavy isotope labeled peptides were used: SLTDLNQDPS [R(13C6;15N4)],    DALRSLTDLNQDPS[R(13C6;15N4)],    DALR[S

(PO3H2)]LTDLNQDPS[R(13C6;15N4)], DALT[S(PO3H2)]LTDLNQD PS[R(13C6;15N4)], and DALTSLTDLNQDPS[R(13C6;15N4)]. Proteolytic digestion and quantitative liquid chromatography–ion mobility separation–mass spectrometry (LC-IMS-MS) was performed as described in the Appendix Supplementary Methods.

### Glycosylation studies using co-immunoprecipitation and Western blotting

For glycosylation studies, co-IP was performed using protein G agarose beads and a custom-made anti-PRG-1 antibody (Trimbuch *et al*, 2009) according to standard procedures. Western blotting was performed using a anti-O-GlcNAc antibody (1:1,000 dilution; CTD110.6, Cell Signaling, USA). For quantification, membranes were stripped and the total amount of PRG-1 was assessed using the above-described anti-PRG-1 antibody. For details, see Appendix Supplementary Methods.

### Immunostaining

Cell labelings were performed using established antibodies and following standard protocols. For details, see Appendix Supplementary Methods.

### Behavioral analyses

All experiments were conducted in accordance with the national laws for the use of animals in research and with the European Communities Council Directive 86/609/EEC and were approved by the local ethical committee (Landesuntersuchungsamt, Rheinland-Pfalz, Germany, #23 177-07/G 12-1-096). Social interaction, PPI, and tail suspension test (before and after acute stress) were conducted as described elsewhere. For details, see Appendix Supplementary Methods.

### Pharmacological intervention

Animals were treated using the *in vivo* potent autotaxin inhibitor PF 8380 (Gierse *et al*, 2010). Animals received a single i.p. injection 3 h prior to PPI. PF8380 concentration in the cerebrospinal fluid (CSF) was measured by mass spectrometry 3 h after injection, yielding a concentration up to 0.42 pmol/μl (with an $IC_{50}$ of 0.1 pmol/μl) (Gierse *et al*, 2010).

### PRG-1 genotype effect on P50 event-related potential measures in humans

Study participants were investigated in the context of the German multicenter study "Genetics of nicotine dependence and neurobiological phenotypes". This study was approved by the ethics committees of each study site's local university and was conducted according to the Declaration of Helsinki and informed consent was obtained from all subjects as described in detail by Lindenberg *et al* (2011) and Quednow *et al* (2012). Subjects were healthy as assessed by a standard medical examination, a drug screen, and routine clinical laboratory tests. They had no current psychiatric disorder according to DSM-IV axis I as assessed by the standardized psychiatric interview (SCID-1) and were not on any medication affecting CNS function.

Across study sites, the study including recordings of EEG was conducted according to the same standard operating procedure.

A detailed description of stimulus presentation (auditory double click paradigm), EEG recording, and signal analysis of the P50 event-related potential (ERP) has been provided in previous publications (Brinkmeyer *et al*, 2011; Quednow *et al*, 2012). The most widely used P50 measure is the S2/S1 ratio (Turetsky *et al*, 2007). However, recent data suggest that individual S1 and S2 amplitudes as well as the S1-S2 difference may exhibit higher heritability (Anokhin *et al*, 2007) and superior test–retest reliability than the S2/S1 ratio (Fuerst *et al*, 2007). In view of this lack of superiority of one particular P50 measure over the others, statistical analyses of *prg-1* genotype effects on P50 ERP measures were performed on S1 and S2 amplitudes, S1-S2 difference, and S2/S1 ratio.

Genotyping was performed by pyrosequencing according to manufacturer's instructions using the PSQ 96 SNP reagent kit (Qiagen, Hilden, Germany) on a PSQ HS96A instrument (Qiagen, Hilden, Germany). Primer sequences and assay conditions are available on request.

### Immunocytochemistry on human tissue

Samples from the somatosensory cortex of a human brain were obtained from routine autopsy in accordance with local ethical committee guidelines. The subject died of non-neurological causes. Tissue was immersion-fixed in 4% PFA over 48 h and snap-frozen on dry ice. Co-localization studies were performed on cryosections using a PRG-1 antibody (custom-made antibody described and characterized by Trimbuch *et al*, 2009 was used in a dilution of 1:500) and antibodies against GAD67 (1:500; mab5406, Chemicon), NeuN (1:1,000; mab377, Chemicon), MAP2 (1:1,000; mab 3418, Chemicon), PSD-95 (1:250, mab1569; Millipore), parvalbumin (1:500; PV235, Swant), and calretinin (1:500; 6B3, Swant). Confocal images were taken on a Leica SP5.

### Statistical analysis

For animal experiments, mice from the same litter or of similar age were chosen. The investigator was blinded for the genotype of the animals. Following experiments, experimental results were analyzed, animals were regenotyped, and corresponding final statistical analyses were performed. Briefly, after assessing for normal distribution (using the Kolmogorov–Smirnov test), comparison between two groups, if not otherwise stated, was performed using a two-tailed unpaired *t*-test for normal distributed data or a Mann–Whitney *U*-test for nonparametric data. Comparison between more than two groups was performed using an one-way ANOVA for parametric data or a Kruskal–Wallis test for nonparametric data. Variances were similar between groups and were estimated using either the *F*-test or the Brown–Forsythe test and the Bartlett's test. *Post hoc* analysis for more than two groups for parametric data was performed using the Bonferroni adjustment for multiple comparisons and Dunn's test for nonparametric data. For behavioral analyses assessing genotype effects and treatment effects, a 2-way ANOVA was used. Statistical analyses of data (including testing for normality using the Kolmogorov–Smirnov test) were performed with GraphPad Prism 5/6 (La Jolla, CA, USA) or with SPSS (Chicago, IL, USA).

**The paper explained**

**Problem**
A mutation (PRG-1$^{R345T}$) in the human PRG-1 gene with a monoallelic frequency of ~0.6% affects 5 million European and US citizens and results in an altered signal integration in the central nervous system. We uncovered the underlying molecular signaling pathway at the synapse which involves bioactive lipids controlling basal glutamatergic release in cortical neurons. This pathway is regulated by PRG-1 which interacts with bioactive phospholipids eventually modulating excitation and signal integration.

**Results**
Applying molecular, electrophysiological, and behavioral analyses in rodents, we show that PRG-1$^{R345T}$ is a loss-of-function mutation, incapable of phospholipid interaction and internalization. Monoallelic loss of PRG-1 in rodents led to loss of somatosensory filter function and altered resilience during stress-related behaviors, which are both regarded as an endophenotype for psychiatric disorders. Electrophysiological assessment of monoallelic human PRG-1$^{R345T}$ carriers revealed a similar reduced sensory filter function.

**Impact**
Using pharmacological intervention into phospholipid signaling, we were able to rescue the altered cortical somatosensory filter function in an animal model with monoallelic PRG-1 deficiency pointing to a new therapeutic rationale for psychiatric disorders.

For statistical analysis of human data, complete datasets including P50 ERP measures and *prg-1* genotype were available from six study centers for $N = 1,811$ subjects ($N = 12$ monoallelic carriers of the *prg-1* variant versus $N = 1,799$ wt/wt controls). There were no deviations from Hardy–Weinberg equilibrium (exact test: $P = 1.0$). Major and minor allele frequencies (99.669 and 0.331%, respectively) were consistent with the frequencies reported previously by the NHLBI Exome Sequencing Project. Key sociodemographic sample characteristics are provided in Appendix Table S1 (Lehrl, 1999).

To avoid confounders such as age, smoking, and study site on P50 ERP measures, we performed a stratified analysis of *prg-1* genotype effects. For each EEG measure, three strata were determined by predicting for each individual the expected value from a regression model, incorporating age, smoking, and study site, and dividing individuals into distinct, roughly equally sized strata, based on visual inspection of the histograms of predicted values. This model for stratum assignment mainly uses information from the wt/wt controls to control for confounding. In a sensitivity analysis excluding the carriers from model fitting, the results did not change. Subsequently, the standardized mean difference between *prg-1* wt/wt and monoallelic R345T carriers was determined in each stratum and aggregated over the strata using inverse variance weighting, as commonly used in fixed effect meta-analysis (Cooper & Hedges, 1994). The assumption of normal distribution, required to obtain p-values corresponding to the aggregated differences, was checked by Q-Q plots. For the S1 and S2 combined effect, the standardized mean difference was obtained from a linear mixed-effects model incorporating a per-subject random effect. For all other measures, standardized mean differences were directly obtained from mean values and standard deviations, which were calculated separately for the carriers and wt/wt controls, that is, the uneven size of the two groups does not skew the results.

Similarly, the linear mixed-effects models also are not affected by the uneven group sizes. These analyses were performed in the statistical environment R (Team, 2011) using the package "meta" (version 2.1-0).

**Expanded View** for this article is available online.

## Acknowledgements

The authors thank Melanie Pfeifer and Nicolai Schmarowski for excellent technical assistance. The studies were funded by the European Research Council (ERC-AG "LiPsyD") and the Deutsche Forschungsgemeinschaft (DFG) within the CRC1080 (to RN, JV, IT, JH, HJL, and AS).

## Author contributions

JV and RN have designed the experiments and have written the paper. JV, SK, HJL, JWY, JH, KR, OT, SG, AS, AM, GW, and RN have contributed to the experimental design and have supervised the experiments and the data analysis. AJM, IT, NF, ST, and UD have performed mass spectrometry analysis. JC, YL, XL, JB, CT, SK, PU, JWY, SR, NS, UD, ST, LQ, KL, OT, HB, NF, IT, AJM, PN, and MRT have been involved in the different experiments. KR and GH have performed the behavioral studies.

## Conflict of interest

The authors declare that they have no conflict of interest.

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
