## [Review Process File · EMBO Molecular Medicine]

Molecular cause and functional impact of altered synaptic lipid signaling due to a prg-1 gene SNP

Johannes Vogt, Jenq-Wei Yang, Arian Mobascher, Jin Cheng, Yunbo Li, Xingfeng Liu, Jan Baumgart, Carine Thalman, Sergei Kirischuk, Petr Unichenko, Guilherme Horta, Konstantin Radyushkin, Albrecht Stroh, Sebastian Richers, Nassim Sahragard, Ute Distler, Stefan Tenzer, Lianyong Qiao, Klaus Lieb, Oliver Tüscher, Harald Binder, Nerea Ferreiros, Irmgard Tegeder, Andrew J. Morris, Sergiu Groppa, Peter Nürnberg, Mohammad R. Toliat, Georg Winterer, Heiko J. Luhmann, Jisen Huai, Robert Nitsch

Corresponding author: Johannes Vogt, University Medical Center, Johannes Gutenberg-University

Review timeline:

Transfer date:	25 July 2015
Editorial Decision:	12 August 2015
Revision received:	12 November 2015
Accepted:	12 November 2015

Transaction Report:

Editor: Céline Carret

1st Editorial Decision

12 August 2015

Thank you for the submission of your manuscript to EMBO Molecular Medicine. We have now received the enclosed reports from three referees (see below).

As you will see all three reviewers are globally supportive of publication and we would appreciate if you could experimentally address referee 3 point 1 regarding the provision of appropriate controls. Other than that, nothing else is required, which is unusual and clearly indicates a very interesting and well performed study.

Please submit your revised manuscript within 3 months.

I look forward to reading a new revised version of your manuscript as soon as possible.

***** Reviewer's comments *****

Referee #1 (Remarks):

This is an excellent work following on previous studies of the group and providing now a translational relevance. They show that a mutation prg-1 gene cause alter synaptic lipid signaling.

This is highly relevant for human disease and extends the medical relevance of this synaptic mechanism.

Referee #2 (Remarks):

This is a novel, potentially clinical relevant study of genetic variation in a gene that is probably important in synaptic signalling.

Referee #3 (Comments on Novelty/Model System):

This manuscript has been well-designed and performed. The testing of this point mutation is clearly novel and essential. It becomes encouraging that more and more lipid-related signaling has been found related to brain disorders.

Referee #3 (Remarks):

1) In most of experiments, if the authors were able to provide gfp-only-expressing group that would be better. This may not be required for every experiment, but, for some key tests such as figure 2 it may be more convincing.

2) The behavior section is relatively weak. However, given this study is not focusing on behavioral tests, it may be acceptable.

1st Revision - authors' response

12 November 2015

***** Reviewer's comments *****

Referee #1 (Remarks):

This is an excellent work following on previous studies of the group and providing now a translational relevance. They show that a mutation in the *prg-1* gene causes altered synaptic lipid signaling. This is highly relevant for human disease and extends the medical relevance of this synaptic mechanism.

Referee #2 (Remarks):

This is a novel, potentially clinical relevant study of genetic variation in a gene that is probably important in synaptic signalling.

Referee #3 (Comments on Novelty/Model System):

This manuscript has been well-designed and performed. The testing of this point mutation is clearly novel and essential. It becomes encouraging that more and more lipid-related signaling has been found related to brain disorders.

Referee #3 (Remarks):

1) In most of experiments, if the authors were able to provide gfp-only-expressing group that would be better. This may not be required for every experiment, but, for some key tests such as figure 2 it may be more convincing.

2) The behavior section is relatively weak. However, given this study is not focusing on behavioral tests, it may be acceptable.

EMM-2015-05677

We thank the Editor and the reviewers for their careful and the very positive evaluation of our manuscript. We have addressed the point mentioned by reviewer 3 and provide now the requested controls for the key experiment shown in Figure 2i.

Referee #3 (Remarks):

1) In most of experiments, if the authors were able to provide gfp-only-expressing group that would be better. This may not be required for every experiment, but, for some key tests such as figure 2 it may be more convincing.

We thank the reviewer for this important remark. To prove that the transfection procedure did not introduce bias when analyzing the TF-LPA uptake of PRG-1^{-/-}, PRG-1^{-/-}//PRG-1^{R345T}-transfected and PRG-1^{-/-}//PRG-1^{wild type}-transfected neurons, we have performed a new set of experiments comparing TF-LPA uptake of non-transfected PRG-1^{-/-} and GFP-only-transfected PRG-1^{-/-} neurons (PRG-1^{-/-}//GFP-transfected). Analysis of these neurons (shown in the Extended View Figure EV1) revealed no difference in the TF-LPA uptake of non-transfected and GFP-transfected neurons. This is in line with the results in wild type neurons (depicted in Figure 2J) showing that GFP-only-transfection did not alter TF-LPA uptake capability of wild type neurons.